# FedDefuse: Mitigating Strategic Model Poisoning for Federated Learning via Divide-and-Compute Driven Composite Behavioral Analysis

## Abstract

Federated Learning (FL) enables collaborative model training across distributed clients without sharing local data, but it is highly vulnerable to strategic model poisoning, where adversaries dominate participation rounds and may selectively launch arbitrary attacks under non-i.i.d. data. Existing defenses, often relying on single-perspective behavioral heuristics, fail to reliably distinguish and suppress malicious behaviors due to the erased distinctions between benign and malicious updates. In this paper, we propose FedDefuse, a principled defense framework built upon a novel composite behavioral pattern that judiciously fuses two complementary indicators, intra-client recoverability and inter-client similarity, in a divide-and-compute manner. FedDefuse first divides the uploaded model updates into two candidate clusters based on their recoverability, which quantifies how faithfully each update can be reproduced through simulated local training. It then identifies benign updates as those exhibiting higher similarity scores to provisional benign clusters in the frequency domain. This design allows FedDefuse to effectively suppress adversarial contributions in the global aggregation without sacrificing benign ones. Extensive experiments demonstrate that FedDefuse significantly outperforms state-of-the-art defenses under strategic model poisoning scenarios, achieving considerable improvements in terms of both detection and model accuracy across diverse settings.

## 1 Introduction

Security is a critical concern in distributed machine learning (ML), where numerous untrusted clients collaboratively train a global model (Liu et al., 2022). As a privacy-aware distributed paradigm, federated learning (FL) proceeds round by round and allows clients to perform local training and communicate solely model updates—rather than raw data—with the central server (Li et al., 2020) in each round. While FL mitigates data exposure risks, it simultaneously creates a broad attack surface for model poisoning, where malicious clients upload crafted and erroneous model updates to compromise global model integrity(Fang et al., 2020; Wu et al., 2022). Recent studies have identified a wide spectrum of attack vectors (Bagdasaryan et al., 2020; Wu et al., 2020), which can severely decelerate model training, degrade global model accuracy or introduce subtle, targeted vulnerabilities, posing substantial threats to the reliability of downstream ML tasks (Lyu et al., 2022).

This paper investigates a practical model poisoning attack scenario in FL, termed *strategic model poisoning*, where: 1) clients hold highly non-i.i.d. data owing to the heterogeneous nature of local data sources and operational environments (Zhao et al., 2018); 2) malicious clients frequently constitute the majority of participants in training rounds, attributed either to a large proportion of attackers in the client pool or the partial participation mechanism, which randomly selects a small subset of clients for synchronization under constrained communication budgets (Zhu et al., 2023); 3) adversaries possess complete knowledge of others and are capable of launching arbitrary attack types (Shejwalkar & Houmansadr, 2021), and more critically, strategically selecting attack timing, i.e., when to launch attacks, to remain covert and maximize their impact (Fang et al., 2020). Such scenario closely mirrors practical FL deployments, where client heterogeneity, limited communication resources, and adaptive attackers coexist, demanding urgent attention.

While a range of defense strategies have been proposed (Zhang et al., 2022; Yan et al., 2023), they fail to account for *strategic model poisoning*. These methods primarily operate at the global aggregation stage of FL, following a *detection-and-aggregation* procedure in each round: the server identifies potentially malicious clients using carefully designed behavioral patterns in local model updates that benign participants adhere to while adversaries deviate from (Fung et al., 2020; Mu et al., 2024; Zhao et al., 2022; Xie et al., 2024). Suspicious clients are then excluded or suppressed from global aggregation based on detection outcomes(Shen et al., 2016), mitigating their adverse impacts on the global model. However, these defenses critically depend on single-perspective behavioral heuristics which expect benign participants to share "similar" characteristics in a single indictor, such as model update magnitude (Fung et al., 2020), reconstructed data (Zhao et al., 2022), local optimization trajectories(Xie et al., 2024), and the like. These patterns are insufficient to consistently capture the nuanced differences between benign and strategically crafted malicious updates under *strategic model poisoning*. In particular, the natural divergence caused by non-i.i.d. data reduces the cohesion among benign updates, while adversarial ones—often dominant—are either chaotically scattered or deliberately crafted to resemble benign updates(Xie et al., 2020). This leads to substantial overlap and erratic model update distributions, rendering such behavioral patterns ineffective and thus making *strategic model poisoning* fall beyond the protective scope of existing defenses.

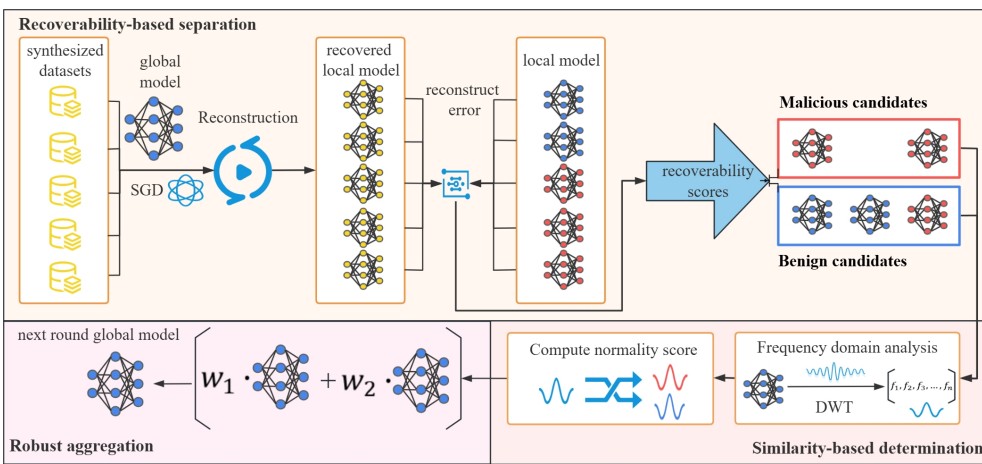

Figure 1: Overview of the proposed FedDefuse framework.

To cope with *strategic model poisoning*, this paper presents FedDefuse, a principled defense framework based on a novel composite behavioral pattern. Instead of relying on uni-dimensional heuristics, the proposed behavioral pattern judiciously fuses two crucial indicators, intra-client model update recoverability and inter-client frequency-domain benign similarity, to establish a discriminative criterion for effectively distinguishing malicious from benign updates. This fusion yields a key behavioral insight: *a benign model update tends to share low-frequency characteristics with model updates that can be reliably and efficiently recovered via simulated local training, whereas a malicious update often appears similar to updates that exhibit poor recoverability*. We remark that the pattern exploits the combined strengths of these two indicators and persists in *strategic model poisoning*, which enables FedDefuse to expose strategically crafted manipulations that bypass traditional defenses.

FedDefuse implements the composite behavioral pattern at the server side in a divide-and-compute pipeline that sequentially examines these two indicators, before consolidating them into a unified discriminator. As shown in Figure 1, this pipeline comprises two key stages: 1) recoverability-based division, where model updates are divided into two candidate clusters based on their reconstruction fidelity via simulated local training; and 2) similarity-based evaluation, where each model update is assessed with respect to its spectral alignment with provisional benign and malicious clusters in the frequency domain. FedDefuse assigns a normality score to the model update of each participant, with higher scores indicating stronger alignment with the behavioral pattern of benign clients. The global model is protected by aggregating these model updates according to their normality scores to mitigate the adversarial contributions. The contribution is summarized as follows:

- We propose a novel composite behavioral pattern for characterizing benign versus malicious model updates in FL under *strategic model poisoning*. It jointly integrates intra-client recoverability and inter-client frequency-domain similarity, complementing each other to

reveal both local training consistency and global alignment. To the best of our knowledge, this is the first composite behavioral pattern specifically designed for model poisoning attacks in the FL literature.

- We develop a robust defense framework, FedDefuse, that operationalizes this composite behavioral pattern by fusing the recoverability and similarity metrics in a divide-and-compute manner. It partitions model updates based on recoverability analysis, followed by assigning normality scores via frequency-domain similarity, enabling accurate suppression of adversarial updates while preserving benign ones through robust aggregation.

- We conduct extensive experiments to demonstrate the superiority of FedDefuse in defending against *strategic model poisoning*. Our evaluation spans different non-i.i.d. degrees, varying participant and malicious client ratios, multiple attack types and adaptive strategies. We remark that FedDefuse consistently outperforms state-of-the-art defenses, achieving near-100% detection accuracy and maintaining the convergence and high utility of global model.

## 2 PROBLEM STATEMENT

### 2.1 FEDERATED LEARNING

We consider the FL system with a central server and $N$ clients which collaboratively learn a shared ML model $\boldsymbol{\theta} \in \mathbb{R}^d$. Assume each client $i$ processes a dataset $\mathcal{D}_i$, the FL process aim to train $\boldsymbol{\theta}$ such that it performs the best across all local datasets $\mathcal{D}_i, i \in [N]$. Given the loss function $f(\boldsymbol{\theta}; \mathbf{x})$ which captures the prediction error of the model $\boldsymbol{\theta}$ on a data sample $\mathbf{x}$, the problem of FL is formulated as

$$\min_{\boldsymbol{\theta} \in \mathbb{R}^d} F(\boldsymbol{\theta}) \triangleq \frac{1}{N} \sum_{i=1}^{N} F_i(\boldsymbol{\theta}; \mathcal{D}_i) \triangleq \mathbb{E}_{\mathbf{x} \in \mathcal{D}_i}[f(\boldsymbol{\theta}; \mathbf{x})], \quad (1)$$

where $F_i(\boldsymbol{\theta}; \mathcal{D}_i)$ is the (possibly non-convex) local cost function of client $i$.

Following the widely-adopted FedAvg algorithm, FL is performed by repeating the following three primary steps in a round-by-round fashion until convergence. In particular, at any round $r$: 1) A set of clients $\mathcal{S}^r$ ($|\mathcal{S}^r| \leq N$) is randomly sampled from $[N]$ and the server broadcasts the global model $\boldsymbol{\theta}^r$ to them; 2) Each client $i \in \mathcal{S}^r$ updates local model $\boldsymbol{\theta}_i^{r+1}$ through $\tau \geq 1$ consecutive steps of stochastic gradient decent (SGD) on the dataset $\mathcal{D}_i$ with $\boldsymbol{\theta}^r$ as the initial, i.e., $\boldsymbol{\theta}_i^{r,0} = \boldsymbol{\theta}^r$,

$$\boldsymbol{\theta}_i^{r,t+1} = \boldsymbol{\theta}_i^{r,t} - \eta g_i(\boldsymbol{\theta}_i^{r,t}), t = 0, \ldots, \tau - 1, \quad (2)$$

where $\boldsymbol{\theta}_i^{r+1} = \boldsymbol{\theta}_i^{r,\tau}$, $g_i(\boldsymbol{\theta}_i^{r,t})$ is the stochastic gradient (SG), and $\eta > 0$ is the local learning rate. Then, each update $\boldsymbol{\theta}_i^{r+1} - \boldsymbol{\theta}^r$ is uploaded to the server; 3) The server aggregates the received local models $\boldsymbol{\theta}_i^{r+1}$'s to update the global model $\boldsymbol{\theta}^{r+1}$, i.e., $\boldsymbol{\theta}^{r+1} = \boldsymbol{\theta}^{r+1} + \frac{1}{|\mathcal{S}^r|} \sum_{i \in \mathcal{S}^r} (\boldsymbol{\theta}_i^{r+1} - \boldsymbol{\theta}^r)$.

### 2.2 ATTACK MODEL

We consider the attack scenario of *strategic model poisoning*, where a subset of $M$ malicious clients ($M < N$) from $[N]$, denoted by $\mathcal{M}$, infiltrate the FL system to launch model poisoning attacks for compromising the integrity of the global model $\boldsymbol{\theta}$. It is worth-noting that the local datasets are heterogeneous, i.e., $\mathcal{D}_i, i \in [N]$ follow non-i.i.d. data distributions. The number of malicious clients $M$ and their identities are totally unknown to the server. At any round $r$ of the FL process, the set of participants $\mathcal{S}^r$ is assumed to contain malicious clients (i.e., $\mathcal{S}^r \cap \mathcal{M} \neq \emptyset$), and more importantly, malicious clients may frequently constitute the majority of $\mathcal{S}^r$, i.e., $|\mathcal{S}^r \cap \mathcal{M}| > \frac{|\mathcal{S}^r|}{2}$.

Once completing local training, each participant uploads its model update to the server. Unlike benign participants, malicious ones may instead submit a deliberately poisoned model update. We assume that each malicious client can strategically decide whether to launch an attack in a given round, based on its objectives or coordination with other adversaries. Let $\mathcal{A}^r \subseteq \mathcal{S}^r \cap \mathcal{M}$ denote the subset of malicious participants that actively conduct attacks at round $r$. Then, the model update $\Delta_i^r$ uploaded by client $i \in \mathcal{A}^r$ is given by

$$\Delta_i^r = \begin{cases} *, & \text{if } i \in \mathcal{A}^r, \\ \boldsymbol{\theta}_i^{r+1} - \boldsymbol{\theta}^r, & \text{if } i \in \mathcal{S}^r \setminus \mathcal{A}^r, \end{cases} \quad (3)$$

where $*$ denotes a malicious update crafted to realize arbitrary attack types. Malicious clients can arbitrarily and strategically manipulate their uploaded updates, thereby encompassing a wide spectrum of poisoning behaviors, from untargeted noise injection to sophisticated targeted attacks.

---

**Algorithm 1** Recoverability Analysis

---

1: **Input:** $\boldsymbol{\theta}^r, \boldsymbol{\Delta}^r, \tilde{\mathcal{D}}_i^r, \tilde{\mathcal{D}}_i^s, \eta_l^i, \forall i \in \mathcal{S}^r, \tilde{\tau}, r, I, K$.
2: **for** client $i \in \mathcal{S}^r$ in parallel **do**
3:     **if** $r \bmod I = 0$ **then**
4:         Compute $\tilde{\mathcal{D}}_i^{s,r} = \mathrm{DLG}(F_i, \boldsymbol{\theta}^r, \Delta_i^r)$ based on the loss in (5c).
5:     **end if**
6:     **for** $k = 0$ to $K - 1$ **do**
7:         Set $\tilde{\boldsymbol{\theta}}^0 = \boldsymbol{\theta}^r$, $\tilde{\boldsymbol{\theta}} = \tilde{\boldsymbol{\theta}}^{\tilde{\tau}}$ and $\tilde{\mathcal{D}}_i^r = \tilde{\mathcal{D}}_i^{r-1}$
8:         Conduct $\tilde{\tau}$ consecutive steps of SGD, i.e., $\tilde{\boldsymbol{\theta}}^{l+1} = \tilde{\boldsymbol{\theta}}^l - \eta_l^i \nabla F_i(\tilde{\boldsymbol{\theta}}^l; \tilde{\mathcal{D}}_i^r), l = 0, \ldots, \tilde{\tau} - 1$.
9:         Update $\tilde{\mathcal{D}}_i^r, \eta_l^i$ using a Adam optimizer based on the loss $\frac{\|\tilde{\boldsymbol{\theta}} - \boldsymbol{\theta}^r - \Delta_i^r\|_2^2}{\|\Delta_i^r\|_2^2} + \lambda \mathrm{KL}(\tilde{\mathcal{D}}_i^r, \mathcal{D}_i^{s,r})$.
10:     **end for**
11: **end for**
12: Keep updated $\tilde{\mathcal{D}}_i^{s,r}$ and $\tilde{\mathcal{D}}_i^r$ at the server side for next-round computation.
13: Compute $\boldsymbol{\Delta}_b^r$ and $\boldsymbol{\Delta}_m^r$ by clustering the set $\{R(\Delta_i^r)\}$.
14: **Output:** $\boldsymbol{\Delta}_b^r, \boldsymbol{\Delta}_m^r, \{R(\Delta_i^r)\}$.

---

### 2.3 DEFENSE OBJECTIVE

Our objective is to defend against *strategic model poisoning* attacks by maximally suppressing the influence of poisoned model updates on the utility of global model. To this end, we aim to develop a discriminative behavioral pattern that clearly distinguishes malicious updates from benign ones, upon which a defense mechanism can be established to reliably identify and filter out adversarial updates from global aggregation in each round of FL.

Formally, let $\boldsymbol{\Delta}^r \triangleq \{\Delta_i^r\}_{i \in \mathcal{S}^r}$ denote the set of uploaded model updates at round $r$. We define a score function $\Phi(\cdot; \boldsymbol{\Delta}^r) : \mathbb{R}^d \to \mathbb{R}$, which assigns a normality score to each input update in the context of the entire set $\boldsymbol{\Delta}^r$. The function $\Phi(\cdot; \boldsymbol{\Delta}^r)$ is instantiated based on the proposed behavioral pattern, such that the resulting normality score quantitatively reflects the likelihood of the input model update being benign by measuring its alignment with the expected behavioral characteristics. Note that the function $\Phi(\cdot; \boldsymbol{\Delta}^r)$ relies solely on the current update set $\boldsymbol{\Delta}^r$, which is is essential as malicious participants may not launch attacks persistently across rounds, rendering historical information unreliable and even misleading (Zhang et al., 2022). Based on $\Phi$, model updates in $\boldsymbol{\Delta}^r$ are selectively filtered or reweighted during global aggregation to produce $\boldsymbol{\theta}^{r+1}$.

## 3 METHODOLOGY

This section introduces FedDefuse, a principled defense framework against *strategic model poisoning*. The essence is a novel composite behavioral pattern that unifies two complementary indicators, intra-client model update recoverability and inter-client frequency-domain benign similarity. Following a divide-and-compute strategy, FedDefuse sequentially fuses these indicators during global aggregation. The process begins with recoverability analysis, which divides model updates into provisionally benign and suspicious groups, followed by frequency-domain similarity evaluation, which assigns normality scores to model updates based on their alignment with the established provisional clusters. Finally, model updates are aggregated in a robustness-aware manner guided by these normality scores.

### 3.1 SEPARATING MODEL UPDATES VIA RECOVERING LOCAL OPTIMIZATION TRAJECTORY

FedDefuse first partitions the set of model updates $\Delta^r$ into two disjoint subsets, $\Delta_b^r$ and $\Delta_m^r$, such that $\Delta_b^r$ contains as many benign updates as possible while ensuring that the vast majority of model updates within it are indeed benign, and $\Delta_m^r$ includes the rest. This partition is guided by a key behavioral distinction: benign updates, derived from authentic local training, can be reliably recovered via multi-step SGD from the global model and local data, whereas malicious updates—crafted to disrupt training—deviate from genuine optimization trajectories and thus cannot be reliably and easily recovered in the same way. We remark that this recoverability-based criterion persists under *strategic model poisoning*: since each model update is evaluated independently based on its consistency with local training dynamics, without cross-client comparisons, it remains effective under highly non-i.i.d. data or when malicious clients dominate the participant set.

To realize recoverabilty analysis, we synthesize a small surrogate dataset $\mathcal{D}_i^s$ for each client $i$ using gradient inversion (Geiping et al., 2020). Following FedInv (Zhao et al., 2022), the surrogate dataset, containing a small number of representative data samples (e.g. $|\mathcal{D}_i^s| = 10$), is optimized such that the gradient computed on them closely matches the normalized model update $\boldsymbol{\Delta}_i$:

$$\min_{\mathcal{D}_i^s} \|\nabla_{\boldsymbol{\theta}} F_i(\boldsymbol{\theta}; \mathcal{D}_i^s) - \frac{\Delta_i}{\tau}\|^2. \tag{4}$$

By this way, though $\mathcal{D}_i^s$ is not identical to the original local data for privacy protection, it sufficiently captures the underlying data distribution and enables the server to simulate the client's local optimization trajectory for assessing the recoverability of its model update.

Then, the recoverability of model update is quantitatively measured by the minimum discrepancy between it and those reconstructed through simulated local training over the entire space of plausible configurations. In particular, at round $r$, we denote $R(\Delta_i^r)$ as the recoverability score for the update $\Delta_i^r, i \in \mathcal{S}^r$, which equals to the optimal objective value of the following optimization problem:

$$\min_{\tilde{\mathcal{D}}_i^r, \eta_l} \frac{\|\tilde{\boldsymbol{\theta}} - \boldsymbol{\theta}^r - \Delta_i^r\|^2}{\|\Delta_i^r\|_2^2} + \lambda \mathrm{KL}(\tilde{\mathcal{D}}_i^r, \mathcal{D}_i^s), \tag{5a}$$

$$\text{s.t.} \quad \tilde{\boldsymbol{\theta}} = \mathcal{U}(\tilde{\mathcal{D}}_i^r, \boldsymbol{\theta}^r, \eta_l, \tilde{\tau}), \tag{5b}$$

$$\mathcal{D}_i^s = \arg\min_{\mathcal{D}} \|\nabla_{\boldsymbol{\theta}} F_i(\boldsymbol{\theta}^r; \mathcal{D}) - \frac{\Delta_i^r}{\tau}\|^2, \tag{5c}$$

where $\tilde{\mathcal{D}}_i^r$ is a candidate dataset used for simulating local training process; $\lambda > 0$ is a penalty parameter; $\mathcal{U}(\tilde{\mathcal{D}}_i^r, \boldsymbol{\theta}^r, \eta_l, \tilde{\tau})$ denotes the local training process on $\tilde{\mathcal{D}}_i^r$, with the global model $\boldsymbol{\theta}^r$, the learning rate $\eta_l$ and a predefined number of local steps $\tilde{\tau}$; $\mathrm{KL}(\tilde{\mathcal{D}}_i^r, \mathcal{D}_i^s)$ denotes the sum of the Kullback-Leibler divergence among all samples of $\tilde{\mathcal{D}}_i^r$ and $\mathcal{D}_i^s$. Here, the penalty term in (5a) ensures that, instead of directly exploiting the surrogate dataset $\mathcal{D}_i^s$, we optimize $\tilde{\mathcal{D}}_i^r$ within the neighborhood of $\mathcal{D}_i^s$ for simulating realistic local training that is consistent with the received model update. This design accounts for the optimization error of solving the subproblem (5c) to obtain $\mathcal{D}_i^s$, yielding a refined one that more accurately reflects the true data distribution and training dynamics underlying the received model update.

Notably, one can see from problem (5) that, the optimization of the surrogate dataset $\mathcal{D}_i^s$ is completely decoupled from that of $\tilde{\mathcal{D}}_i^r$ and $\eta_l$, forming a bi-level optimization structure. It thus can be tackled by solving (5c) for $\mathcal{D}_i^s$, followed by iteratively performing two nested training loops for optimizing $\tilde{\mathcal{D}}_i^r$ and $\eta_l$. Algorithm 1 outlines the detailed procedure. At first, the DLG method in (Zhao et al., 2020) is adopted to solve (5c), resulting in the update of surrogate dataset $\tilde{\mathcal{D}}_i^{s,r}$ every $I$ rounds (Step 3–7). Note that we update the surrogate dataset periodically to balance computational efficiency and the fidelity of reconstructed data. Next, for each active client $i$, the two nested training loops (Step 8–16) optimize $\tilde{\mathcal{D}}_i^r$ and $\eta_l$ to pursue the recoverability score $R(\Delta_i^r)$. Specifically, the inner loop (Step 10–12) simulates genuine consecutive SGD steps to obtain the reconstructed model update $\tilde{\boldsymbol{\theta}}$ based on the current guess $\tilde{\mathcal{D}}_i^r$ and $\eta_l^i$, while the outer loop seeks to refine $\tilde{\mathcal{D}}_i^r$ and $\eta_l$ using Adam such that the resulting $\tilde{\boldsymbol{\theta}}$ aligns closely with the update $\Delta_i^r$ and remains faithful to the prior $\mathcal{D}_i^s$. Finally, the model updates in $\Delta^r$ are clustered into two groups based on $\{R(\Delta_i^r)\}$, where $\boldsymbol{\Delta}_b^r$ contains updates with lower recoverability scores, and the rest are included by $\boldsymbol{\Delta}_m^r$.

## 3.2 COMPUTING NORMALITY SCORE VIA SIMILARITY ASSESSMENT

With the two sets $\boldsymbol{\Delta}_b^r$ and $\boldsymbol{\Delta}_m^r$, FedDefuse instantiates the score function $\Phi(\cdot; \boldsymbol{\Delta}^r)$ by assigning each update $\Delta_i^r, i \in \mathcal{S}^r$, a normality score that accurately quantifies its likelihood of being benign. Assuming that $\boldsymbol{\Delta}_b^r$ predominantly consists of benign updates, while $\boldsymbol{\Delta}_m^r$ mostly contains anomalous ones, the normality score is expected to be positively (resp. negatively) correlated with the distance to $\boldsymbol{\Delta}_b^r$ (resp. $\boldsymbol{\Delta}_m^r$), thereby reflecting how well a model update aligns with benign behavioral patterns. Drawing on this intuition, we propose a distance-based scoring mechanism that relies on the relative closeness of each update to these two reference sets, as shown in Algorithm 2.

**Benign and malicious distance.** The proposed scoring mechanism relies on two complementary metrics: benign distance and malicious distance. For a given update $\Delta_i^r$ at round $r$, we define its

benign distance $d_b(\Delta_i^r; \mathbf{\Delta}_b^r)$ and malicious distance $d_m(\Delta_i^r; \mathbf{\Delta}_m^r)$ as its distance to the representative elements of $\mathbf{\Delta}_b^r$ and $\mathbf{\Delta}_m^r$, respectively. As the distinction between benign and malicious model updates is highly obscured in the parameter space under *strategic model poisoning*, we instead transform them into the frequency domain and focus on their low-frequency components. This is motivated by prior findings(Xu et al., 2019; 2020) that the energy of model parameters is concentrated in low-frequency components, and the model tends to learn from low to high frequencies during training. Therefore, malicious attacks often introduce distinct patterns that alter these low-frequency characteristics within benign model updates(Fereidooni et al., 2024), making the low-frequency spectrum more informative for identifying underlying behavioral differences.

Following this insight, we adopt Discrete Wavelet Transform (DWT) (Zhang, 2019) to extract compact low-frequency representations from model updates, and denote it by $\text{DWT}(\Delta)$ for any model update $\Delta$. To compute the benign and malicious distance, we evaluate the distance between the low-frequency representation of a given model update and the representative elements of $\mathbf{\Delta}_b^r$ and $\mathbf{\Delta}_m^r$. Formally, the benign distance $d_b(\Delta_i^r)$ and malicious distance $d_m(\Delta_i^r)$ are defined as

$$d_b(\Delta_i^r; \mathbf{\Delta}_b^r) = \|\text{DWT}(\Delta_i^r) - \text{DWT}(\bar{\Delta}_b^r)\|, \tag{6}$$

$$d_m(\Delta_i^r; \mathbf{\Delta}_m^r) = \sum_{\Delta' \in \bar{\mathbf{\Delta}}_m^r} \|\text{DWT}(\Delta_i^r) - \text{DWT}(\Delta')\|, \tag{7}$$

where $\bar{\Delta}_b^r$ and $\bar{\mathbf{\Delta}}_m^r$ respectively denote the representative elements of $\mathbf{\Delta}_b^r$ and $\mathbf{\Delta}_m^r$. It is worth-noting that, as benign model updates constitute the majority of $\mathbf{\Delta}_b^r$, $\bar{\Delta}_b^r$ can be interpreted as a single benign prototype, obtained as the geometric median of all elements in $\mathbf{\Delta}_b^r$ in the frequency domain:

$$\bar{\Delta}_b^r = \arg\min_{\Delta} \sum_{\Delta' \in \mathbf{\Delta}_b^r} \|\text{DWT}(\Delta) - \text{DWT}(\Delta')\|. \tag{8}$$

Since malicious model updates are typically scattered and exhibit high variance, $\bar{\mathbf{\Delta}}_m^r$ is defined as a set of $K$ representative elements within $\mathbf{\Delta}_m^r$ ($K \le |\mathbf{\Delta}_m^r|$) with the highest recoverability scores, i.e.,

$$\bar{\mathbf{\Delta}}_m^r = \arg_{\mathbf{\Delta} \subseteq \mathbf{\Delta}_m^r, |\mathbf{\Delta}|=K} \max \sum_{\Delta' \in \mathbf{\Delta}} R(\Delta'). \tag{9}$$

By this way, $\bar{\Delta}_b^r$ (resp. $\bar{\mathbf{\Delta}}_m^r$) is more likely to capture the dominant benign (resp. malicious) behavioral patterns. Accordingly, $d_b(\Delta_i^r; \mathbf{\Delta}_b^r)$ quantifies how well $\Delta_i^r$ aligns with the benign behavioral pattern, whereas $d_m(\Delta_i^r; \mathbf{\Delta}_m^r)$ measures how closely it aligns with the representative malicious behaviors.

**Computing normality score.** A benign model update should align with benign behavioral patterns while deviating significantly from malicious ones. Motivated by this, we define the normality score of a model update $\Delta_i^r$ at round $r$ as

$$\Phi(\Delta_i^r; \mathbf{\Delta}^r) = \alpha \tilde{d}_m(\Delta_i^r; \mathbf{\Delta}_m^r) - (1-\alpha)\tilde{d}_b(\Delta_i^r; \mathbf{\Delta}_b^r), \tag{10}$$

where $\alpha \in [0,1]$ is a tunable parameter that balances the contributions of the two components. Here, to ensure compatibility and numerical stability, both distance scores $\tilde{d}_b(\cdot)$ and $\tilde{d}_m(\cdot)$ are rescaled to the range $[0,1]$ via Min-Max normalization. The deflated scores are respectively denoted as $\tilde{d}_m(\Delta_i^r; \mathbf{\Delta}_m^r)$ and $\tilde{d}_b(\Delta_i^r; \mathbf{\Delta}_b^r)$. Naturally, the score function $\Phi(\Delta_i^r; \mathbf{\Delta}^r)$ outputs a normality score for the input model update $\Delta_i^r$, with higher values indicating a greater likelihood of it being benign.

### 3.3 Normality Score Guided Robust Aggregation

For each round, FedDefuse ends with a robust global aggregation procedure, with the aim of adhering two key principles: 1) ensuring the inclusion of all benign updates to preserve model utility; and 2) excluding the influence of adversarial updates to maintain robustness.

As summarized in Algorithm 2, we propose to employ a two-stage clustering strategy to identify and retain benign model updates based on their normality scores, rather than relying on simple thresholding schemes. First, we apply the clustering algorithm DBSCAN to the frequency-domain benign similarity space (Step 9 of Algorithm 2) to verify whether the update distribution exhibits uniformly benign behavior. If all updates are deemed benign, we aggregate without exclusion. Otherwise, we perform KMeans clustering (Step 13 of Algorithm 2) over the set of participants

---

**Algorithm 2** Normality Score Construction and Model Update filtering

---

1: **Input:** $d_b(\Delta_i^r), d_m(\Delta_i^r), \Delta_i^r, i \in \mathcal{S}^r$.
2: Compute $\bar{\Delta}_i^r$ via Equation (8), and compute $\bar{\boldsymbol{\Delta}}_m^r$ via Equation (9)
3: **for** client i $\in \mathcal{S}^r$ **do**
4:    Compute $d_b(\Delta_i^r)$ via (6), compute $d_m(\Delta_i^r)$ via (7), and compute $\Phi(\Delta_i^r)$ via (10)
5: **end for**
6: D-labels = DBSCAN($\{d_b(\Delta_i^r), i \in \mathcal{S}^r\}$), $\mathcal{B}^r = \mathcal{S}^r$
7: **if** not only one distinct element in D-labels **then**
8:    $labels$ = KMeans($\{\Phi(\Delta_i^r), i \in \mathcal{S}^r\}$)
9:    $high\_label = \arg_{a \in labels} \max \sum_{label(i)=a} \Phi(\Delta_i^r)$
10:    $\mathcal{B}^r = \{i, label(i) = high\_label\}$
11: **end if**
12: **Output:** $\mathcal{B}^r$.

---

to divide candidate model updates into two clusters, and select the cluster with a higher average normality score as the benign candidate set $\mathcal{B}^r$. Then, the global model is updated using the weighted average of model updates in $\mathcal{B}^r$, i.e.,

$$\boldsymbol{\theta}^{r+1} = \boldsymbol{\theta}^r + \eta \cdot \frac{1}{|\mathcal{B}^r|} \sum_{i \in \mathcal{B}^r} \frac{|\mathcal{D}_i|}{|\mathcal{D}|} \Delta_i^r. \tag{11}$$

This adaptive clustering-based filtering allows FedDefuse to maintain high global model accuracy while defending against adaptive and stealthy model poisoning attacks.

## 4 EXPERIMENTS

### 4.1 EXPERIMENT SETTINGS

**Datasets and DNN models.** Two popular benchmark datasets are considered for the experiments: the CIFAR-10(Krizhevsky, 2009) and Fashion-MNIST(Xiao et al., 2017) datasets, and two different deep neural networks(DNNs) are adopted for the two datasets, respectively. To simulate non-i.i.d. data, we employ the partition scheme in (Acar et al., 2021) to distribute the training samples of each dataset to $N = 100$ clients. It works by allocating data samples of a few labels according to the Dirichlet distribution. The Dirichlet hyperparameter $\beta$ determines the degree of data heterogeneity, and a smaller $\beta$ corresponds to a higher degree of heterogeneity. The value of $\beta$ is 0.1 by default.

**Attacks types and patterns.** Four attack types of model poisoning are considered for evaluation, including Sign-flipping attack(Wu et al., 2020), Scaling attack(Bagdasaryan et al., 2020), Gaussian attack(Ma et al., 2022), and Fang attack(Fang et al., 2020). We denote by $\bar{\mathcal{A}}$ the maximum ratio of malicious participants in each round, and a big $\bar{\mathcal{A}}$ means that a high ratio of adversaries among all participants would happen. The value of $\bar{\mathcal{A}}$ is 0.6 by default. We also instantiate three attack patterns to simulate attack timing selection under *strategic model poisoning*: 1) Pattern I, where all malicious participants will attack at every round; 2) Pattern II, where malicious participants will attack only in the first 50 communication rounds; and 3) Pattern III, where each malicious participant launch attacks with probability $p \in (0, 1)$ per round. The value of $p$ is 0.2 by default.

**BaseLines and evaluation metrics.** The state-of-art defense algorithms chosen to compare against FedDefuse includes Auror(Shen et al., 2016), FoolsGold(Fung et al., 2020), FedDMC(Mu et al., 2024), FedCPA(Han et al., 2023), FedFreq(Fereidooni et al., 2024), RFA(MINSKER, 2015), Krum(Blanchard et al., 2017), Median(Xie et al., 2018) and FedREDefense(Xie et al., 2024). We consider three evaluation metrics: Test Accuracy (TACC), Defense Accuracy (DACC), Malicious Rate (MR) and Aattack Success Rate (ASR). The metric TACC measures the utility of global model, while DACC, MR and ASR measures the detection accuracy of malicious model updates.

### 4.2 PERFORMANCE EVALUATION UNDER ATTACK PATTERN I

Figure 2 illustrates the convergence performance comparison between FedDefuse and baseline algorithms with respect to communication rounds under different attack types on the CIFAR-10 dataset. As depicted, FedDefuse maintains a DACC approaching 100% and a MR of 0% across the

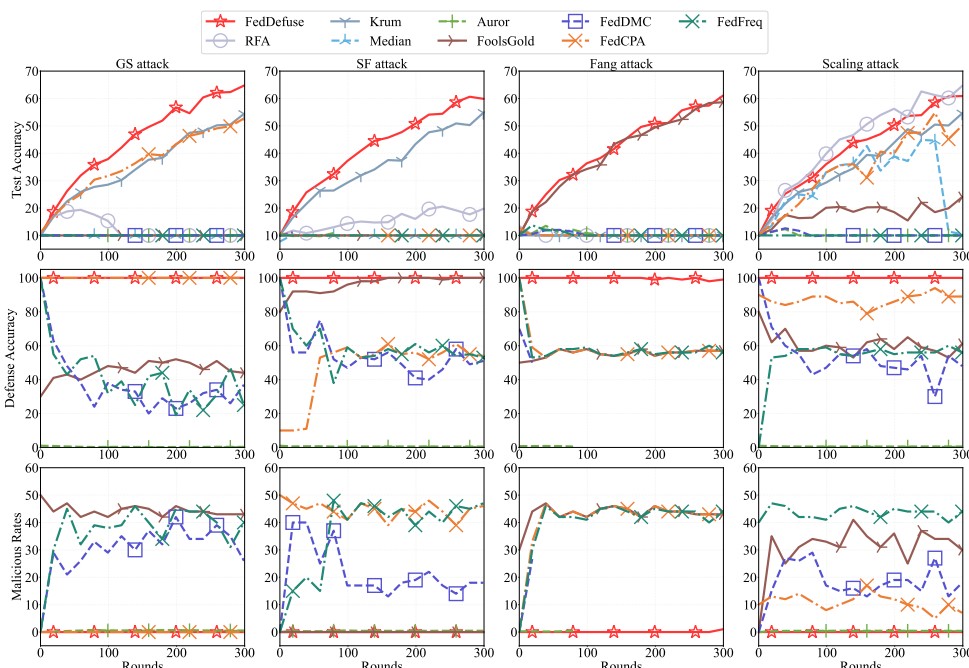

Figure 2: Performance comparison between FedDefuse and baselines under different attacks on the CIFAR-10 dataset. Among them, the first row of subfigures use TACC as the metric, the second row of subfigures use DACC, and the third row use MR as the metric.

Table 1: Performance evaluation results on the CIFAR10 dataset. The symbol "-" means that the algorithm crashes.

| Algorithm | GS-attack | | | SF-attack | | | Fang-attack | | | Scaling-attack | | | |
|---|---|---|---|---|---|---|---|---|---|---|---|---|---|
| | TACC | DACC | MR | TACC | DACC | MR | TACC | DACC | MR | TACC | DACC | MR | ASR |
| No Defense | 10.0 | 55.3 | 44.7 | 10.0 | 55.3 | 44.7 | 10.0 | 55.3 | 44.7 | 10.0 | 55.3 | 44.7 | 90.0 |
| RFA | 10.0 | - | - | 19.8 | - | - | 10.0 | - | - | 64.4 | - | - | 89.8 |
| Krum | 55.6 | - | - | 56.0 | - | - | 10.0 | - | - | 55.6 | - | - | 6.6 |
| Median | 10.1 | - | - | 10.0 | - | - | 10.0 | - | - | 10.5 | - | - | 61.7 |
| Auror | 10.0 | 36.4 | 55.6 | 10.0 | 78.2 | 26.6 | 10.0 | 58.5 | 41.5 | 10.0 | 58.1 | 42.5 | 90.0 |
| FoolsGold | 10.0 | 44.6 | 44.4 | 10.0 | 55.1 | 44.1 | 58.4 | 97.9 | 0 | 23.6 | 58.6 | 32.4 | 89.3 |
| FedDMC | 10.0 | 31.4 | 33.6 | 10.0 | 66.3 | 13.8 | 10.0 | 52.9 | 41.2 | 10.0 | 53.7 | 19.0 | 90.0 |
| FedCPA | 54.1 | 100 | 0 | 10.0 | 56.5 | 43.0 | 10.0 | 40.0 | 43.7 | 49.3 | 86.3 | 10.8 | 89.6 |
| FedFreq | 10.0 | 38.6 | 39.7 | 10.0 | 56.3 | 43.0 | 10.0 | 58.1 | 37.7 | 10.0 | 55.1 | 44.7 | 90.0 |
| **FedDefuse** | **65.0** | **100** | **0** | **59.4** | **99.7** | **0** | **60.7** | **100** | **0** | **60.5** | **100** | **0** | **4.2** |

entire training trajectory so it achieves a faster convergence than other algorithms. This outcome highlights FedDefuse's effectiveness and robustness in mitigating model poisoning attacks throughout the FL process. In contrast, other baseline algorithms exhibit significant fluctuations in these metrics, underscoring their instability when confronting model poisoning attacks.

Table 1 presents the final defense performance of FedDefuse and other state-of-the-art algorithms, when defending against various attacks on the CIFAR-10 dataset. Specifically, when confronted with SF attack, GS attack, and Fang attack, FedDefuse consistently outperforms other algorithms across the three key metrics: TACC, DACC, and MR. We remark that, under Scaling attack, FedDefuse demonstrates remarkable robustness, achieving 100% DACC and the lowest ASR.

### 4.3 PERFORMANCE EVALUATION UNDER ATTACK PATTERN II AND III

Figure 3 presents a comparative analysis of FedDefuse, FedREDefense(Xie et al., 2024) and best defense settings on CIFAR-10 under attack Pattern I and II. FedDefuse implements a dynamic client

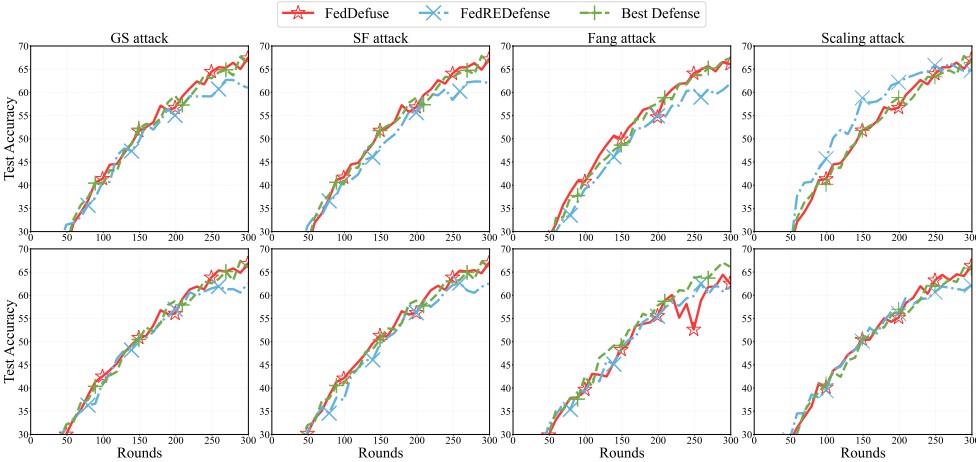

Figure 3: Performance comparison under model poisoning attack Pattern II and III. The first row of subfigures correspond to Pattern II, and the second row of figures correspond to Pattern III.

identification mechanism, actively screening participants each training round, whereas FedREDefense adopts a static exclusion strategy, removing only pre-identified adversaries at the outset. Across all four attack types, FedDefuse consistently outperforms FedREDefense and achieves performance close to the best defense settings under both attack patterns. By contrast, FedREDefense exhibits a decline in final model accuracy. This highlights the superiority of FedDefuse over FedREDefense, demonstrating its enhanced resilience to *strategic model poisoning* scenarios.

Table 2 summarize the detailed TACC achieved by FedDefuse and other baseline algorithms under attack Pattern II and III. We can observe that selectively launching attacks does severely compromise the integrity of the global model while simultaneously increasing the difficulty of reliable defense due to their stealthy and adaptive nature. FedDefuse consistently maintains highest TACC, demonstrating its strong resilience against such strategic adversaries.

Table 2: Test accuracy (TACC) under various attacks on CIFAR-10.

| Attack Pattern | Algorithm | GS-attack | SF-attack | Fang-attack | Scaling-attack |
|---|---|---|---|---|---|
| Pattern II | No Defense | 10.0 | 10.0 | 10.0 | 10.0 |
| | Best Defense | 66.73 | 67.16 | 66.37 | 67.16 |
| | FedREDefense | 60.48 | 63.34 | 60.87 | 64.72 |
| | **FedDefuse** | **67.62** | **67.32** | **66.25** | **67.25** |
| Pattern III | No Defense | 10.0 | 10.0 | 10.0 | 10.0 |
| | Best Defense | 67.02 | 67.17 | 66.44 | 66.46 |
| | FedREDefense | 61.92 | 61.88 | 62.67 | 62.87 |
| | **FedDefuse** | **67.01** | **67.47** | **64.0** | **66.39** |

## 5 CONCLUSION

This paper studies a practical and challenging threat of *strategic model poisoning* in FL. To address it, we propose FedDefuse, a principled defense framework based on a novel composite behavioral pattern. The pattern judiciously fuses intra-client recoverability and inter-client frequency-domain similarity in a divide-and-compute manner, enabling reliable discrimination between benign and malicious behaviors in model updates. Built upon it, FedDefuse computes the normality score for each model update by evaluating its alignment wit the proposed behavioral pattern, and offers a robust global aggregation scheme guided by normality scores to eliminate adversarial contributions to the global model. Extensive experiments show that FedDefuse consistently achieves superior performance, obtaining perfect detection in most rounds and an average DACC exceeding 99%, which significantly outperforms state-of-the-art baselines under *strategic model poisoning*.

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

## A    RELATED WORKS

**Similarity based detection.** This line of work identifies malicious clients by measuring the similarity of their updates to others, under the assumption that poisoned updates deviate from benign ones in certain metric spaces. Early methods rely on statistical distances or clustering. For example, FoolsGold(Fung et al., 2020) detects colluding adversaries via cosine similarity, while Auror(Shen et al., 2016) applies PCA and KMeans to isolate outliers. FedDMC(Mu et al., 2024) improves clustering with a binary tree approach, yet both degrade under non-i.i.d. data or adversary dominance. Beyond parameter space, FedInv(Zhao et al., 2022) reconstructs dummy data from updates and measures Wasserstein distance, but its effectiveness depends on accurate inversion. More recent works exploit structured representations: FreqFed(Fereidooni et al., 2024) applies DCT and density-based clustering, and FedCPA(Han et al., 2023) evaluates consistency of critical parameters. Despite these advances, most methods assume a benign-majority distribution, limiting robustness against strategic model poisoning.

**Optimization-based detection.** Another direction assumes benign updates should align with global optimization objectives or consistent local trajectories. FLTrust(Cao et al., 2021) derives a reference update from a trusted dataset, but this contradicts FL's privacy principle. FLDetector(Zhang et al., 2022) leverages temporal consistency, predicting current updates from historical ones, yet non-i.i.d. data weakens this assumption. FedREDefense(Xie et al., 2024) introduces distilled local knowledge and uses reconstruction errors as indicators, showing improved robustness under high adversary ratios. However, these methods often rely on stable attack patterns, assuming adversaries act persistently and can be permanently excluded. Such assumptions fail under strategic model poisoning, where attackers adaptively disguise their behaviors.

**Robust aggregation.** Instead of explicit detection, robust aggregation aims to mitigate poisoned updates by designing outlier-resilient rules. DeFL(Yan et al., 2023) detects statistical anomalies via gradient norm variations and down-weights suspicious updates during vulnerable training phases. Several aggregation rules directly operate in the parameter space to mitigate adversarial influence. For instance, Trimmed Mean(Yin et al., 2018) removes extreme values in each dimension before averaging, while Median(Xie et al., 2018) computes the coordinate-wise median. Krum and its extension Multi-Krum(Blanchard et al., 2017) select updates that are closest to their neighbors, thereby reducing the impact of outliers. RFA(MINSKER, 2015) further computes the geometric median, reducing sensitivity to outliers. While effective under i.i.d. data and minority adversaries, these rules degrade when malicious clients exceed half or when benign/adversarial updates become entangled. Consequently, their assumptions are incompatible with strategic model poisoning, motivating the need for more adaptive defenses.

In summary, all of the above defenses rely on single-perspective behavioral heuristics and thus are fundamentally constrained by unrealistic assumptions that are incompatible with *strategic model poisoning*. As a result, these methods cause dramatic performance degradation, either failing to reliably identify malicious clients or overly suppressing benign updates. This motivates the urgent need of proposing a defense framework effectively addressing *strategic model poisoning*, which is our key focus in this work.

## B    ADDITIONAL EXPERIMENT DETAILS

We evaluate the effectiveness of FedDefuse in defending against multiple model poisoning attacks by conducting experiments with various experimental settings. The comparison results against multiple state-of-the-art algorithms demonstrate the FedDefuse has outstanding advantages.

## B.1 EXPERIMENTS SETTINGS

**Datasets and models.** Two datasets are used to conduct the experiments: the popular CI-FAR10(Krizhevsky, 2009) and Fashion-MNIST(Xiao et al., 2017) datasets. Two different deep neural networks (DNNs) are adopted for the two datasets. Specifically, the CIFAR-10 DNN model has three fully connected layers, while the Fashion-MNIST model has two fully connected layers.

To simulate non-i.i.d. data, we employ the scheme described in the (Acar et al., 2021), using the Dirichlet distribution to model the distribution of non-i.i.d. data within clients. In the experimental setup, the hyperparameter $\beta$ determines the degree of data heterogeneity, and a smaller $\beta$ corresponds to a higher degree of heterogeneity.

**Attacks.** The detailed specifications of the attacks employed to conduct the experiments reported in this paper are presented below.

- Sign-flipping attack(Wu et al., 2020): Attackers deliberately flip the sign of their model updates before sending them to the central server. In particular, for every malicious worker uploads $\Delta_w^r = -k\hat{\Delta}_w^r$ to the server instead of its true gradient $w \in \mathcal{B}$ with $\hat{\Delta}_w^r$ after training.

- Scaling attack(Bagdasaryan et al., 2020): Attackers strategically inject triggers into their local datasets and scale model weights within the detector's allowable range, thus avoiding backdoor dilution by the average parameters of the server. $\Delta_w^r = s * \hat{\Delta}_w^r$, where $s$ is the scaling coefficient.

- Gaussian attack(Ma et al., 2022): Attackers replace the local model based on Gaussian noise of the mean and variance of the local model parameters. $\Delta_w^r = \mu + N(0,1) * \sigma$, where $\mu$ and $\sigma$ are, respectively, the average value and variance of all parameters of the model.

- Fang attack(Fang et al., 2020): Attackers target Krum and Trimmed Mean/Median aggregation rules with specific strategies, leveraging the latter's transfer capabilities to maximize divergence between attacked model updates and benign aggregates.

**Implementation of the attack pattern.** Three attack patterns are conducted to compare the defense performance of all algorithms. Prior to each experimental round, we first sample the attackers according to the preset malicious number parameter $\bar{\mathcal{A}}$. During any attack round, every selected attacker independently commit a malicious update to the server with the configured probability p, following the specified attack pattern.

**BaseLines.** We selected three categories of baseline defense algorithms to compare with the proposed FedDefuse. The first category includes Auror(Shen et al., 2016), FoolsGold(Fung et al., 2020), FedDMC(Mu et al., 2024), FedCPA(Han et al., 2023), and FedFreq(Fereidooni et al., 2024), most of which are the state-of-art defense algorithms. These methods are only based on clustering via different kinds of distance similarity. The second category includes RFA(MINSKER, 2015), Krum(Blanchard et al., 2017), and Median(Xie et al., 2018). They are very popular defense algorithms that focus on mitigating the impact of attacks by enhancing the aggregation rules. The third category includes FedREDefense(Xie et al., 2024), which is based on the recoverability of model updates. To validate effectiveness, we also conduct a comparison with no defense senario in FL.

- No Defense(McMahan et al., 2017): A baseline setting where the server aggregates local updates through standard federated averaging. This approach is computationally efficient but leaves the system fully exposed to poisoning, backdoor, and sybil attacks, especially under non-i.i.d. data distributions

- Auror(Shen et al., 2016): A defense that applies dimensionality reduction to client updates and then performs KMeans clustering to separate benign and malicious participants. By exploiting the observation that poisoned updates often form distinct clusters, Auror can effectively filter adversarial contributions while maintaining model utility.

- FoolsGold (Fung et al., 2020): A similarity-based defense that leverages PCA and cosine similarity to track the update patterns of clients. Clients exhibiting highly correlated updates are suspected of being sybils and are down-weighted through adaptive learning rate adjustments. This method requires no additional assumptions about the number of attackers and is effective against targeted poisoning.

- FedDMC(Mu et al., 2024): A defense combining PCA-based dimensionality reduction with binary tree clustering enhanced by noise injection. The hierarchical clustering structure improves the separation of benign and malicious updates, while the noise component prevents overfitting to adversarial patterns. A self-ensemble correction mechanism further stabilizes detection across rounds.

- FedCPA(Han et al., 2023): A parameter-centric defense that identifies malicious clients by monitoring changes in critical parameters rather than relying solely on distance metrics. Since benign clients tend to preserve consistent sets of important parameters, deviations in these critical dimensions serve as reliable indicators of poisoning, even under heterogeneous data.

- FedFreq(Fereidooni et al., 2024): A frequency-domain approach that applies the discrete cosine transform (DCT) to model updates and compares their low-frequency components. These components capture the stable, task-relevant structure of updates, making the method robust against adversarial perturbations that typically manifest in high-frequency regions.

- FedREDefense(Xie et al., 2024): A defense that evaluates the recoverability of client updates. Benign updates can be reconstructed with low error using distilled knowledge, while poisoned updates exhibit significantly higher reconstruction error. This property enables reliable filtering of adversaries under both i.i.d. and non-i.i.d. conditions.

- RFA(MINSKER, 2015): A robust aggregation method that replaces the arithmetic mean with the geometric median of client updates. By minimizing the influence of outliers, RFA provides theoretical guarantees of robustness and convergence, making it effective even when a substantial fraction of clients are adversarial.

- Multi_Krum (Blanchard et al., 2017): A outlier-resilient aggregation rule that selects multiple client updates with the smallest summed Euclidean distances to their nearest neighbors and averages them. This design ensures that updates consistent with the majority are preserved, while adversarial outliers are excluded.

- Median(Xie et al., 2018): A coordinate-wise robust aggregation strategy that computes the median across clients for each parameter dimension. This simple yet effective method mitigates the impact of extreme outliers for federated learning.

**Evaluation Metrics.** In order to verify that our defense algorithm can achieve the defense objectives, we use the following experimental metrics under different experimental setups to evaluate the strengths and weaknesses of our defense algorithm against the comparison algorithms.

- Test Accuracy (TACC): the metric to measure the accuracy of the global model on the testing dataset. This metric directly reflect the performance of FL training.

- Defense Accuracy (DACC): the metric used to measure the average proportion of correct predictions out of the total number of samples, across all rounds. This metric emphasizes the identification rate of both benign and malicious clients, but may not directly reflect the effectiveness of defense algorithms against malicious clients.

- Malicious Rates (MR): the metric used to measure the average proportion of malicious clients that are actually identified as benign across all rounds.

- Attack Success Ratio (ASR): the metric used to measure the probability of successfully triggering the backdoors added by Scaling attack, the lower the metric better.

**Algorithms settings.** For FedDefuse, we utilize a IDLG(Zhu et al., 2019) methods with 300 iterations in the server side to construct $\mathcal{D}^s$ for clients, and we set the parameter $\eta_l = 0.01, I = 100, \lambda = 0.01, K = 5$ and $\tilde{\tau} = 16$. For FedREDefense, we set Adam optimizer for image to optimize, and the other parameters remain unchanged. For FedFreq, we implemented it according to its algorithm descriptions. For other algorithms, the parameters are tuned according to the original papers.

Table 3: Performance of different defense algorithms under various attacks on FMNIST datasets. All experiments are conducted under the settings: $\beta = 0.1, \bar{\mathcal{A}} = 60\%$. TACC(%), DACC(%), MR(%), ASR(%) are shown in this table.

| Dataset | Algorithm | GS-attack | | | SF-attack | | | Fang-attack | | | Scaling-attack | | | |
|---|---|---|---|---|---|---|---|---|---|---|---|---|---|---|
| | | TACC | DACC | MR | TACC | DACC | MR | TACC | DACC | MR | TACC | DACC | MR | ASR |
| FMNIST | No Defense | 20.2 | 55.3 | 44.7 | 9.8 | 55.3 | 44.7 | 18.6 | 55.3 | 44.7 | 35.5 | 55.3 | 44.7 | 90.2 |
| | RFA | 31.0 | - | - | 9.8 | - | - | 59.6 | - | - | 94.4 | - | - | 90.2 |
| | Krum | 92.5 | - | - | 9.8 | - | - | 92.3 | - | - | 90.0 | - | - | 0.6 |
| | Median | 13.2 | - | - | 9.8 | - | - | 17.1 | - | - | 77.6 | - | - | 90.2 |
| | Auror | 22.1 | 75.2 | 25.2 | 31.2 | 69.3 | 30.4 | 20.0 | 78.6 | 25.1 | 90.6 | 74.8 | 30.3 | 90.2 |
| | FoolsGold | 20.2 | 49.1 | 44.7 | 94.1 | 95.9 | 0 | 9.8 | 54.3 | 38.5 | 53.0 | 66.4 | 5.5 | 90.2 |
| | FedDMC | 28.9 | 62.8 | 31.8 | 40.2 | 64.4 | 30.9 | 9.8 | 52.9 | 20.9 | 18.0 | 70.2 | 19.6 | 88.4 |
| | FedCPA | 19.2 | 82.6 | 9.9 | 9.8 | 55.6 | 44.3 | 9.8 | 55.2 | 43.4 | 9.8 | 92.3 | 7.6 | 90.2 |
| | FedFreq | 38.4 | 71.3 | 14.3 | 9.8 | 51.7 | 28.1 | 9.8 | 64.2 | 28.8 | 43.4 | 79.3 | 3.9 | 90.2 |
| | **FedDefuse** | **94.4** | **99.4** | **0.03** | **94.4** | **99.4** | **0.03** | **94.4** | **99.7** | **0.3** | **94.5** | **100** | **0** | **0.5** |

## B.2 EVALUATION

### B.2.1 ADDITIONAL RESULTS OF PERFORMANCE COMPARED WITH BASELINES UNDER STRATEGIC ATTACKS PATTERN I

As shown in Table 3, with $\bar{\mathcal{A}}$ is set at 60%, most comparative algorithms struggle to maintain high DACC across diverse attack types, even some of them collapsed. In contrast, FedDefuse consistently achieves a DACC approaching 99% against all evaluated attack methods. Notably, when confronting Scaling attacks employing backdoor attack strategies, FedDefuse suppresses the ASR below 1% on FMNIST datasets. When malicious clients are predominant, clustering algorithms based on distance alone are unable to separate benign clients groups and incorrectly take malicious clients into the aggregation. However, FedDefuse utilize the recoverability based separation to determine the benign group firstly to address the problem.

These experimental results not only demonstrate FedDefuse's superior defense efficacy against diverse attacks across two distinct datasets (CIFAR-10 and FMNIST), but also highlight its exceptional stability and robustness throughout the FL process.

### B.2.2 PERFORMANCE UNDER DIFFERENT NON-I.I.D. SETTINGS

Figure 4 presents the comparative results of FedDefuse and other algorithms under increasing degrees of data heterogeneity on the CIFAR-10 dataset. The data heterogeneity levels are systematically set at $\beta = 0.1, 0.5$, and 2, with a fixed $\bar{\mathcal{A}}$ of 40%. As illustrated, FedDefuse consistently outperforms other algorithms or performs best in terms of TACC for the final trained model, even in the presence of two distinct attack methods and across different degrees of data heterogeneity. Specifically, when $\beta = 2$, FedDefuse achieves a TACC exceeding 80% under both attack scenarios, while at $\beta = 0.1$, it still maintains a TACC above 65%.

Notably, as the data heterogeneity intensifies (manifested by a decreasing $\beta$ value), the performance gap between FedDefuse and the best of other algorithms becomes more pronounced. Specifically, when $\beta = 0.1$, the gaps are more than 5% and 15% under the GS and SF attacks, respectively, which further validates the robustness and adaptability of FedDefuse in heterogeneous FL environments.

### B.2.3 PERFORMANCE COMPARISON UNDER DIFFERENT MALICIOUS NUMBERS

Figure 5 illustrates the comparative results of FedDefuse and other algorithms under varying $\bar{\mathcal{A}}$ on the CIFAR-10 dataset with $\beta = 0.1$. As depicted, under the two attack scenarios of GS attack and Fang attack, FedDefuse maintains stable and dominant TACC (above 60%) as the model poisoning ratio gradually increases. In contrast, baseline algorithms like FedDMC exhibit significant fluctuations and declines in TACC with rising poisoning ratios. Figure 6 further presents the dynamic performance comparison between FedDefuse and other algorithms throughout the FL training process at a fixed

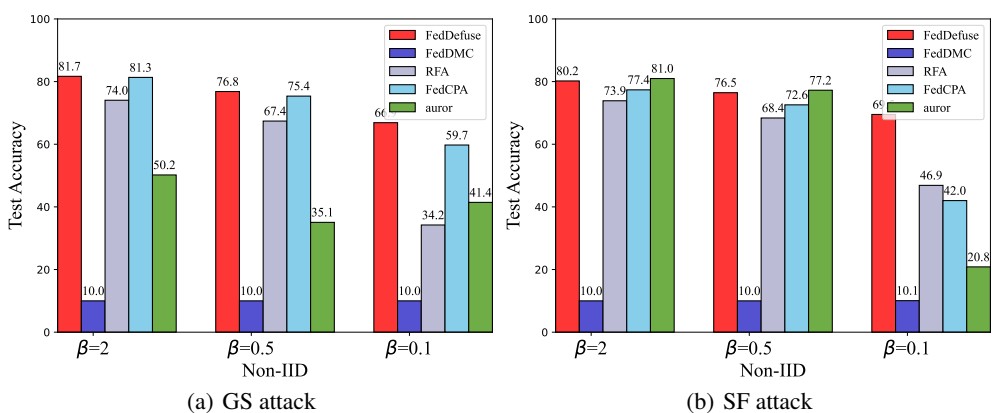

Figure 4: Testing accuracy comparison of different defenses against GS attack and SF attack with various non-i.i.d. degree on CIFAR-10 dataset. $\bar{\mathcal{A}} = 40\%$.

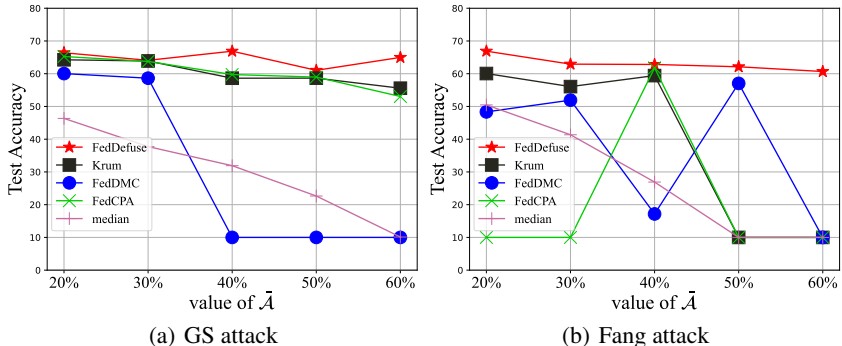

Figure 5: Testing accuracy comparison of different algorithms against GS attack and Fang attack with various $\bar{\mathcal{A}}$ on CIFAR-10 dataset.

$\bar{\mathcal{A}}$ of 40%. Notably, FedDefuse consistently leads in defense metrics across all training rounds, demonstrating its intrinsic robustness without relying on attack-specific adjustments.

Collectively, the results demonstrate that FedDefuse delivers both high efficiency and strong robustness across a broad spectrum of poisoning attacks and attacker ratios.

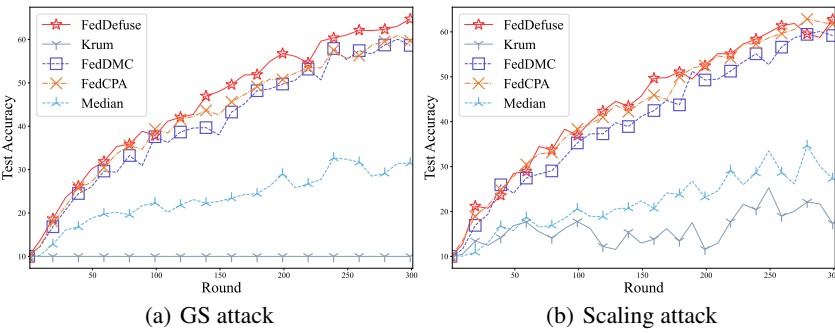

Figure 6: Performance comparison of different defenses against attacks on CIFAR-10 dataset under the settings: $\beta = 0.1$, $\bar{\mathcal{A}} = 40\%$.

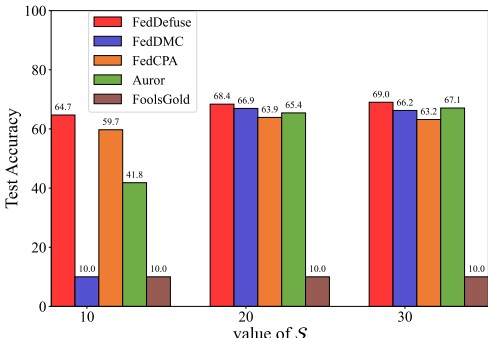

Figure 7: Testing accuracy comparison of different algorithms defense against GS attack with various client participate number on CIFAR-10 datasets.

### B.2.4 PERFORMANCE UNDER DIFFERENT USER SAMPLE RATIOS

Figure 7 illustrates the comparative results of user participation number $\mathcal{S}$ between FedDefuse and other algorithms under GS attack on the CIFAR-10 dataset with $\beta = 0.1$. The total user pool is set to 100 participants (including both benign and malicious users). As depicted in the figure, as the number of users participating in each training round decreases, FedDefuse consistently maintains near-optimal stability and high performance, whereas other algorithms exhibit substantial declines and fluctuations in defense metrics. Notably, when the user participation number drops to 10, FedDefuse demonstrates a more pronounced superiority over baseline algorithms.

This experimental outcome highlights that even in scenarios with limited user engagement (characterized by a small user participation number), FedDefuse can still reliably ensure effective and stable defense against various attacks, underscoring its robustness in low-participation FL environments.

