# OpenReview forum: "FedDefuse: Mitigating Strategic Model Poisoning for Federated Learning via Divide-and-Compute Driven Composite Behavioral Analysis"
_ICLR.cc/2026/Conference — ICLR 2026 Conference Withdrawn Submission_

### Official Review · Reviewer_9qyr · 2025-10-28

**Soundness:** 3
**Presentation:** 2
**Contribution:** 2
**Rating:** 2
**Confidence:** 5

**Summary:**

This paper proposes FedDefuse, a defense framework for federated learning (FL) against strategic model poisoning attacks, where adversaries can dynamically choose when and how to attack. The key idea is to combine two behavioral indicators: (1) Recoverability: how well a model update can be reconstructed via simulated local training using gradient inversion; (2) Frequency-domain similarity: low-frequency feature similarity via wavelet transforms. The authors design a two-stage “divide-and-compute” pipeline: first cluster updates via recoverability, then perform a spectral similarity scoring and weighted aggregation. Extensive experiments on CIFAR-10 and Fashion-MNIST claim superior results compared to 9 existing defenses.

**Strengths:**

1. Proposing a new threat model of strategic poisoning (adaptive and non-persistent adversaries).
2. Combining recoverability and frequency-domain features is an interesting direction beyond single-indicator heuristics.
3. The paper is generally well-written and clearly structured.
4. Experimental results are comprehensive in terms of datasets and baselines.

**Weaknesses:**

1. Practicality of the Threat Model Needs Clarification. The authors need to justify the practicality of the proposed threat model. The so-called strategic model poisoning scenario does not clearly align with either cross-device or cross-silo federated learning settings. It remains unclear whether such a threat is realistic in real-world deployments. The paper should provide concrete examples or empirical evidence demonstrating that this threat model can occur in practical FL environments.

2. Missing Evaluation under Minority-Adversary Scenarios. The paper exclusively evaluates FedDefuse under adversary-majority settings. However, existing literature has shown that even controlling a single malicious client can be sufficient to poison the global model. While it is reasonable to position FedDefuse as a defense targeting majority-controlled adversaries, the authors should also demonstrate its capability to identify a small number of compromised clients among predominantly benign ones.

3. Insufficient Discussion of Relevant Baselines. Recent defense methods such as [1] have also shown strong robustness in highly non-i.i.d. settings with adversaries controlling more than half of the clients. It would be beneficial to include a discussion and experimental comparison with these baselines to clarify FedDefuse’s relative position and contribution.

4. Lack of Evaluation Against Advanced Backdoor Attacks. Advanced backdoor attack methods such as Neurotoxin [2] and AutoAdapt [3] can inject backdoors by modifying only a small subset of model parameters, producing malicious updates that closely resemble benign ones. It is unclear whether FedDefuse can distinguish such subtle poisoning behaviors. The authors are encouraged to evaluate these stronger attacks to validate the generality of their defense.

5. Scalability to Larger Models is Not Demonstrated. The current evaluation uses shallow DNN architectures, which may not reflect realistic federated learning scenarios where deeper models such as ResNet18 or ResNet34 are commonly used. It is important for the authors to demonstrate whether FedDefuse remains effective and computationally feasible on larger and more complex models.

[1] BackdoorIndicator: Leveraging OOD Data for Proactive Backdoor Detection in Federated Learning.

[2] Neurotoxin: Durable Backdoors in Federated Learning.

[3] Automatic adversarial adaption for stealthy poisoning attacks in federated learning.

**Questions:**

1. Coud authors evaluate FedDefuse when only a few (say 1 or 2 or maybe 10\%) malicious updates are uploaded in every global rounds?
2. Could authors justify described threat model and provide concrete examples for the strategic model poisoning?
3. Could authors evaluate FedDefuse against more advanced model poisoning algorithms in comparison with SOTA defenses, and also in larger model?

---

### Official Review · Reviewer_XM6L · 2025-10-30

**Soundness:** 3
**Presentation:** 3
**Contribution:** 2
**Rating:** 4
**Confidence:** 4

**Summary:**

This paper presents a defense framework for federated learning against strategic model poisoning. It combines intra-client recoverability and inter-client frequency-domain similarity within a divide-and-compute framework, enabling more accurate detection of malicious clients and improving robustness in federated learning aggregation.

**Strengths:**

1.	The idea of combining recoverability analysis and frequency-domain similarity provides an interesting and complementary behavioral view for detecting adaptive attacks.
2.	The paper points out that existing FL defenses often target static or non-strategic adversaries. Its exploration of strategic model poisoning extends the discussion to a more practically relevant attack scenario.

**Weaknesses:**

1.	The proposed behavioral indicators, recoverability and frequency-domain similarity, are mainly heuristic. The paper lacks a theoretical justification or formal analysis clarifying why these features correlate with benign behavior, which limits the interpretability and theoretical soundness of the approach.
2.	The integration of recoverability and frequency-domain similarity relies on a manually tuned linear weighting factor $\alpha$ without joint optimization. This fixed combination might limit adaptability under different attack patterns or heterogeneity conditions, and a more adaptive weighting scheme could potentially improve performance.
3.	The divide-and-compute driven pipeline requires additional local reconstruction and frequency analysis, both of which can be computationally demanding. The paper does not discuss the runtime overhead, communication cost, or memory footprint associated with these operations.
4.	The experimental evaluation is limited to two small-scale image datasets and does not cover scenarios with strong data heterogeneity (0.01), which may not fully reflect the method’s performance under stronger heterogeneity or more realistic federated conditions.
5.	The contribution of each component (recoverability vs. frequency analysis) is not quantified, making it difficult to assess which factor drives performance.

**Questions:**

1.	Could the authors provide more theoretical justification for why recoverability and frequency-domain similarity correlate with benign behavior?
2. How sensitive is the method to the manually tuned weighting factor $\alpha$? Have the authors considered an adaptive or jointly optimized weighting scheme?
3. Could the authors quantify the computational, communication, and memory overhead introduced by the divide-and-compute pipeline?
4. It would be valuable to evaluate the method under conditions of stronger data heterogeneity or using larger, more realistic federated datasets.
5. Could an ablation study be added to clarify the individual contributions of recoverability and frequency-domain similarity?

---

### Official Review · Reviewer_4AW6 · 2025-10-30

**Soundness:** 2
**Presentation:** 2
**Contribution:** 2
**Rating:** 2
**Confidence:** 5

**Summary:**

The authors present a defense framework that employs a composite behavioral pattern to enhance the security of federated learning.

**Strengths:**

1. The paper introduces a new approach to strengthen the security of federated learning.

2. Experimental evaluations demonstrate the effectiveness of the proposed approach.

**Weaknesses:**

1. The concept of using data or model reconstruction to mitigate poisoning attacks is not novel.

2. The proposed approach is likely to introduce substantial computational overhead.

3. The paper does not include comparisons with more recent and robust defense methods.

4. The work lacks theoretical guarantees or formal analysis.

**Questions:**

1. The proposed method primarily builds upon the concept of data/model reconstruction, a principle that has already been explored in prior work [a]. However, the paper does not clearly articulate how FedDefuse differs from FedREDefense [a]. Moreover, the authors’ claim that FedREDefense assumes persistent adversaries that can be permanently excluded is inaccurate, as [a] explicitly considers adaptive adversaries.

2. The defense framework requires bi-level optimization and repeated gradient inversion (Algorithm 1) for each client, leading to significant computational overhead. Generating surrogate datasets through DLG/FedInv and performing nested optimization in every training round make the approach infeasible for large-scale FL with many clients or high-dimensional models. Additionally, the paper does not report the computational costs of the proposed method and baselines across all datasets.

3. The reported testing accuracy on CIFAR-10 is notably low. The authors should adopt more advanced architectures such as ResNet for CIFAR-10 and include corresponding computational cost analyses for both the proposed method and comparison baselines under this setting.

4. Federated learning is a well-established research area, yet the defenses evaluated in this paper are outdated. The authors should compare FedDefuse with more recent and robust defense methods, such as [b] and [c].

5. The paper only provides empirical evaluations to demonstrate robustness and lacks any formal theoretical analysis or proof of convergence under adversarial settings.



[a] FedREDefense: Defending against Model Poisoning Attacks for Federated Learning using Model Update Reconstruction Error. In ICML 2024.

[b] FLShield A Validation Based Federated Learning Framework to Defend Against Poisoning Attacks. In IEEE Symposium on Security and Privacy 2024.

[c] Do We Really Need to Design New Byzantine-robust Aggregation Rules. In NDSS 2025.

---

### Official Review · Reviewer_1Eeb · 2025-10-31

**Soundness:** 2
**Presentation:** 3
**Contribution:** 2
**Rating:** 4
**Confidence:** 5

**Summary:**

This paper presents FedDefuse, a defense framework designed to mitigate strategic model poisoning attacks in federated learning (FL). The approach combines two complementary behavioral indicators: intra-client recoverability and inter-client similarity in the frequency domain, through a divide-and-compute architecture. Specifically, the system performs recoverability-based separation of client updates followed by spectral similarity scoring to identify benign and malicious updates for robust aggregation.

**Strengths:**

+ The integration of recoverability and frequency-domain similarity is original and intuitively justified. It offers a dual-perspective characterization of client updates that goes beyond single-metric defenses.
+ The algorithms are well structured, and the divide-and-compute pipeline is clearly explained. The flow from data reconstruction to spectral analysis is logically consistent and easy to follow.
+ The experiments benchmark FedDefuse against multiple poisoning strategies (sign-flipping, scaling, Gaussian, and Fang) and attack patterns (I–III), showing consistent improvements across metrics.
+ The paper is well written and organized, with effective figures and tables that make the method and results easy to understand.

**Weaknesses:**

- The paper does not provide theoretical analysis or formal guarantees explaining why combining recoverability and frequency-domain similarity yields robustness. The mechanism remains empirically supported but theoretically unclear. Recoverability analysis involves iterative gradient inversion and dataset reconstruction, which can be computationally demanding. The paper omits runtime and memory comparisons, leaving scalability uncertain.
- The success of FedDefuse depends on DLG-based surrogate datasets $D_s^i$, which may poorly approximate real client data in deep models. The paper does not evaluate performance under reconstruction noise or low-fidelity data, raising concerns about robustness.
- Experiments rely exclusively on CIFAR-10 and Fashion-MNIST with lightweight CNNs, which are too simple to represent realistic FL environments. These toy-scale setups limit the generality of the conclusions. Since the framework depends on gradient inversion, scalability to deeper architectures (for example, ResNet or Transformer models) is questionable. Evaluation on more complex datasets (for example, CIFAR-100 or Tiny-ImageNet) and deeper models would significantly strengthen the work.
- Important hyperparameters ($\lambda$, $K$, $\tilde{\tau}$, $\alpha$) are fixed without justification or sensitivity analysis. The absence of ablation experiments obscures which component (recoverability or frequency similarity) contributes most to performance gains.
- The paper does not specify random-seed initialization or report variance across multiple runs. All results (for example, DACC, ASR, and ACC) appear to come from single trials, with no mean or standard deviation reported. Given the stochasticity of federated learning, especially under non-IID data, this omission raises serious reproducibility concerns. Without standard deviations or confidence intervals, the claims of stability and “100 % detection accuracy” may be overstated.
- The extremely high detection scores reported without variance measures suggest possible overfitting or selective reporting of results. The lack of statistical evidence weakens the reliability of empirical claims.

**Questions:**

1.What is the computational cost (in time and memory) per FL round? Can FedDefuse scale to hundreds or thousands of clients?

2.How robust is FedDefuse when DLG reconstruction quality degrades? Have you considered using alternative surrogate-data reconstruction or knowledge-distillation approaches?

3.Can FedDefuse resist adversaries that deliberately mimic recoverable trajectories or align spectral components? If not, how could the framework adapt to second-order attacks?

4.How sensitive is performance to the weighting parameter $\alpha$ in Eq. (10)? Please include ablation studies isolating the effects of recoverability versus frequency-domain analysis.

5.Have you evaluated FedDefuse on deeper architectures such as ResNet or Transformer-based models? How would it perform on more complex datasets (for example, CIFAR-100, Tiny-ImageNet)?

6.Were random seeds fixed and shared for reproducibility? How many independent runs were averaged for each reported metric? Can you provide standard deviations or confidence intervals for DACC, ASR, and accuracy results?

---

### Note · Authors · 2026-01-02

I have read and agree with the venue's withdrawal policy on behalf of myself and my co-authors.